# TSQueryBench: LLM-as-a-Judge for Time-Series Explanations

**Preetham Sivalingam** [1]  **Murari Mandal** [2]  **Dhruv Kumar** [3]  **Saurabh Deshpande** [4]

## Abstract

Natural language explanations of time-series data are increasingly produced by foundation models in high-stakes domains, making factual correctness critical. Evaluating such explanations differs fundamentally from standard natural language generation: correctness requires verifying numerical claims against structured data rather than similarity to reference text. While LLM-as-a-Judge has emerged as a scalable paradigm for text evaluation, its applicability to numerically grounded time-series explanations remains unstudied. We introduce **TSQueryBench**, a controlled synthetic benchmark of 500 time-series instances across 10 query types, each paired with correct, partially correct, and incorrect explanations. We evaluate six large language models across four tasks: explanation generation, relative ranking, independent scoring, and multi-anomaly detection. Our central finding is a consistent generation–evaluation asymmetry: models that fail to generate numerically correct explanations nonetheless reliably identify or score correct ones. These results show that rubric-guided LLM evaluation is substantially more reliable than generation for time-series reasoning, supporting LLM judges as scalable evaluators in numerically grounded settings. Code and data: https://github.com/Prxxthxm/TSQueryBench/.

## 1. Introduction

Foundation models are increasingly used to generate natural language explanations of time series signals in domains such as finance, healthcare, industrial monitoring, and climate science (Zhang et al., 2024). As these systems produce fluent interpretations of anomalies, trends, and distributional shifts, ensuring factual grounding in the underlying data becomes critical, as even subtle numerical inaccuracies can lead to incorrect decisions. Reliable evaluation of such data-grounded explanations is therefore both practically important and technically challenging.

Existing evaluation approaches are ill-suited for this setting. Reference-based metrics such as BLEU, ROUGE, and BERTScore measure similarity to a gold reference (Papineni et al., 2002; Lin, 2004; Zhang et al., 2020), but cannot verify numerical correctness. Natural language inference models assess entailment but require reference statements and lack grounding in numerical data (Bowman et al., 2015). Classical time-series methods support tasks such as trend detection and anomaly identification (Hamilton, 1994; Box et al., 2015), but cannot evaluate free-form explanations. Human evaluation is the gold standard but does not scale. LLM-as-a-Judge has emerged as a scalable alternative, with methods such as GPTScore, G-Eval, and MT-Bench showing that LLMs can approximate human judgments under rubric-based prompting (Fu et al., 2023; Liu et al., 2023; Zheng et al., 2023), and reference-free frameworks such as RAGAS enabling evaluation without gold references (Akin et al., 2023). However, whether LLMs can reliably evaluate numerically grounded explanations requiring reasoning over statistical properties and precise quantitative claims remains unstudied.

We address this gap by introducing **TSQueryBench**, a controlled synthetic benchmark of 500 time-series instances across 10 query types, each paired with correct, partially correct, and incorrect explanations derived from ground-truth statistics. We evaluate six LLMs across four tasks: explanation generation, relative ranking, independent scoring, and multi-anomaly detection. Explanation generation serves as a foil, enabling direct comparison between generative and evaluative capabilities. Our contributions are: (1) TSQueryBench, a benchmark with three-level correctness labels across 10 query types; (2) a rubric-guided multi-task evaluation framework; (3) the first systematic study of LLM-as-a-Judge for numerically grounded time-series explanations; and (4) empirical evidence of a robust generation-evaluation asymmetry.

Our results show that evaluation consistently and substantially exceeds generation across query types and model

[1]BITS Pilani, Pilani, India [2]KIIT & Birla AI Labs, Bhubaneshwar, India [3]BITS Pilani & Birla AI Labs, Pilani, India [4]Birla AI Labs, Mumbai, India. Correspondence to: Preetham Sivalingam <f20240610@pilani.bits-pilani.ac.in>.

*Proceedings of the 43rd International Conference on Machine Learning*, Seoul, South Korea. PMLR 306, 2026. Copyright 2026 by the author(s).

families. Even on volatility shift tasks, where generation accuracy is near zero, models correctly rank and score explanations above chance. This asymmetry holds across ranking and independent scoring, indicating it is a robust property of current foundation models. These findings support rubric-guided LLM evaluation as a reliable approach for time-series explanation assessment while highlighting key limitations in data-grounded generation.

## 2. Methodology

### 2.1. Overview

We study the reliability of large language models as judges of natural language explanations grounded in time-series data. Given a time series, a question, and a candidate explanation, the central question is whether an LLM can determine if the explanation is factually consistent with the underlying numerical data. Unlike reference based metrics, our framework conditions the judge directly on the raw time series to verify numerical claims against source data. Ground truth correctness labels are used only for benchmarking and are not available at inference time. We evaluate models across four tasks, as generators and as evaluators in ranking, scoring, and anomaly detection, enabling direct measurement of whether these capabilities are dissociable.

### 2.2. Input Representation and Evaluation Setting

Each instance consists of a univariate time series $T = \{t_1, \ldots, t_n\}$, a question $q$, and an explanation $e$. No reference explanation is provided; correctness is determined by logical and numerical consistency with $T$.

We formulate evaluation as:

$$y = f_\theta(T, q, e), \quad y \in \{0, 1, 2\}$$

where $y = 0$ denotes incorrect, $y = 1$ partially correct with numeric inaccuracies, and $y = 2$ fully correct.

### 2.3. Rubric-Guided Evaluation

Evaluation uses a structured rubric capturing multiple dimensions:

- **Data faithfulness:** correctness of patterns, anomalies, or trends

- **Numeric accuracy:** correctness of magnitudes, indices, and changes

- **Question relevance:** whether the explanation answers the question

- **Logical coherence:** internal consistency of reasoning

- **Unsupported claims:** absence of claims not grounded in the data

The model verifies numerical claims directly against the time series. These dimensions guide a single final classification rather than being scored independently.

### 2.4. Evaluation Tasks

We evaluate four complementary tasks, each designed to probe a specific aspect of generative or evaluative capability.

**Explanation Generation.** Given a time series $T$ and question $q$, the model generates an explanation $\hat{e}$, which is evaluated using the ternary rubric. This task is included as an experimental foil. Generation accuracy for the same models and instances establishes a baseline against which evaluation performance is compared, directly quantifying the generation evaluation gap.

**Relative Ranking.** Given $T$, $q$, and three candidate explanations $\{e_0, e_1, e_2\}$ corresponding to incorrect, partially correct, and correct reasoning, the model selects the best explanation. This is the primary evaluation task, directly testing LLM as a judge capability in a comparative setting.

**Independent Scoring.** Given $T$, $q$, and a single explanation $e$, the model assigns a ternary correctness label without access to alternatives. This evaluates whether rubric guided assessment remains reliable in the absence of comparative context, reflecting a more realistic deployment scenario.

**Multi-Anomaly Detection.** Given a time series $T$, the model identifies all anomalies along with their indices and percentage changes. This task probes direct numerical reasoning without an explanation to evaluate, establishing quantitative grounding as a prerequisite for reliable judgement.

### 2.5. Design Choices and Justification

**Why rubric-guided evaluation?** A rubric enables structured assessment of numerical accuracy and logical consistency beyond surface-level similarity.

**Why condition on raw data rather than reference answers?** Reference based metrics compare to gold text but do not verify numerical correctness. Our framework verifies claims directly against the time series. Ground truth labels are used only for benchmarking and are not available at inference.

**Why multiple evaluation tasks?** Generation tests production capability, ranking and scoring test evaluation ability, and anomaly detection tests direct numerical reasoning. To-

gether, they provide a comprehensive view of model behavior.

## 3. Experimental Setup

### 3.1. Models

We evaluate six large language models: Qwen-3 8B, LLaMA-3.1 8B, Gemma-4 31B, Gemma-3 12B, Claude-3 Haiku, and Deepseek v3.2, spanning variation in parameter scale (8B to 31B), training paradigm (instruction-tuned open-weight versus proprietary), and architectural family (Qwen, LLaMA, Gemma, and Claude). This diversity allows us to test whether observed failure modes are model-specific or reflect systematic limitations of current foundation models on numerical reasoning tasks. Each model is evaluated both as a generator of explanations and as an evaluator in ranking and scoring tasks, enabling direct comparison under a unified setting.

### 3.2. Datasets

We evaluate on **TSQueryBench**, a synthetic benchmark of 500 time-series instances spanning ten query types: linear spike, seasonal drop, structural break, multi-metric consistency, relative extremum, mean shift, volatility shift, trend comparison, temporal ordering, and shape classification. Each query type contains 50 instances, with sequence lengths of 100, 200, 300, and 500 time steps. The structure of query types is loosely inspired by prior time-series QA formulations, though all instances and evaluation tasks are constructed independently.

For each instance, we generate three candidate explanations corresponding to correct, partially correct, and incorrect reasoning using a rule-based pipeline. Faithful explanations are derived from ground-truth statistics (e.g., anomaly location, percentage change, z-score, or distributional properties), partially correct explanations introduce controlled numerical perturbations, and incorrect explanations consist of plausible but factually inconsistent reasoning patterns such as shifted indices or incorrect trends. This yields 1,500 explanation instances for evaluation in the independent scoring and ranking tasks. Full details of the construction pipeline are provided in Appendix C.

The multi-anomaly detection task uses a separate dataset of 100 time-series instances containing between 1 and 10 anomalies each, evenly distributed across sequence lengths of 100, 200, 300, and 500. Representative samples and full evaluation traces are provided in Appendix B.

### 3.3. Evaluation Protocol

All tasks use structured prompts that instruct the model to reason directly over the underlying time series rather than relying on values stated in the explanation.

For explanation generation, model outputs are evaluated by a domain-knowledgeable annotator using the ternary rubric described in Section 2. The annotator is provided with the raw time series, the question, and the model output, and verifies all numerical claims against the data.

For relative ranking, accuracy is measured as the proportion of instances where the correct explanation is selected, with candidate order randomized to eliminate positional bias. For independent scoring, accuracy is computed against ground-truth labels across all candidate explanations. For multi-anomaly detection, performance is evaluated using count accuracy and F1 score, with predicted indices matched to ground truth within a tolerance of $\pm 2$ time steps.

### 3.4. Implementation Details

All experiments are conducted in a zero-shot setting without task-specific fine-tuning. Prompts are standardized across models to ensure comparability, and outputs are constrained to structured JSON formats for deterministic parsing. Fixed decoding settings are used to reduce stochastic variability, and all evaluations are performed on the same set of instances.

## 4. Results and Discussion

### 4.1. Experiment 1: Explanation Generation

We first establish generation performance as the baseline against which evaluation results in Experiments 2 and 3 are contrasted. The gap between the two forms the central empirical finding of this paper.

**Findings.** Overall generation performance varies substantially across query types (see Appendix A.1). While models achieve strong accuracy on certain structured patterns, no model performs consistently well across all tasks.

*Universal failure on Volatility Shift.* All models fail almost completely on Volatility Shift, indicating a systematic inability to reason about changes in variance. This failure persists across model scale and architecture, suggesting a fundamental limitation in current LLM reasoning over higher-order statistical properties.

*Format sensitivity in structured reasoning.* Performance differences on tasks such as Mean Shift reveal strong sensitivity to output structure. Models that produce explicitly structured, rubric-aligned explanations achieve significantly higher accuracy, indicating that correctness is tightly coupled to output format rather than underlying reasoning capability alone.

*Table 1.* Summary of model performance across all four evaluation tasks in TSQueryBench. Best per row in **bold**, second-best underlined.

| Metric | LLaMA 3.1 8B | Gemma 4 31B | Gemma 3 12B | Qwen 3 8B | Haiku 3 | DS v3.2 |
|---|---|---|---|---|---|---|
| Exp 1 Acc | 0.18 | 0.36 | 0.13 | **0.38** | 0.31 | 0.30 |
| Exp 2 Acc | 0.49 | 0.67 | 0.56 | **0.84** | 0.41 | 0.74 |
| Exp 3 Acc | 0.29 | 0.67 | 0.47 | **0.72** | 0.35 | 0.56 |
| Exp 4 Acc | 0.01 | **0.45** | 0.00 | 0.12 | 0.01 | 0.25 |
| Exp 4 F1 | 0.34 | **0.88** | 0.28 | 0.53 | 0.62 | 0.87 |

## 4.2. Experiment 2: Relative Ranking

**Findings.** Ranking accuracy is consistently higher than generation accuracy across query types (Appendix A.2), revealing a clear asymmetry between generative and evaluative capabilities.

*Generation-evaluation asymmetry.* Ranking accuracy consistently and substantially exceeds generation accuracy across all query types and model families (Table 1, Appendix A). Most notably, on Volatility Shift, where all models achieve near-zero generation accuracy, models still select the correct explanation at above-chance rates. This dissociation holds across both weaker and stronger models and is further confirmed by the independent scoring results in Experiment 3, establishing it as a robust finding rather than an artifact of the ranking setting.

*Robustness to input length.* Ranking performance remains relatively stable across time-series lengths, indicating that evaluation tasks are less sensitive to input scale compared to generation.

## 4.3. Experiment 3: Independent Scoring

**Findings.** Independent scoring performance closely matches ranking results (Appendix A.3), further reinforcing the reliability of evaluation relative to generation.

*Evaluation remains strong without comparative context.* The generation-evaluation asymmetry persists when comparative context is removed. Models maintain strong scoring accuracy when evaluating single explanations in isolation, confirming that the asymmetry is not an artifact of the ranking setting. This has practical implications, as an LLM judge can reliably assess individual outputs without requiring comparison against alternatives, which is more representative of real-world deployment.

## 4.4. Experiment 4: Multi-Anomaly Detection

**Findings.** Models show low count accuracy but moderate F1 scores across all sequence lengths (Appendix A.4).

*High sensitivity, low calibration.* Models tend to overes-

timate anomalies rather than miss them, suggesting high sensitivity to deviations but poor calibration in estimating exact counts.

## 4.5. Discussion

Taken together, the results support a qualified affirmative answer to the central question: LLMs can serve as reliable judges of time-series explanations, particularly for tasks involving pattern recognition, trend identification, and structural change detection. Rubric guided prompting enables evaluative accuracy that consistently exceeds generative accuracy across models and query types, suggesting that the primary limitation lies in generation rather than underlying numerical understanding. However, two important boundaries on this conclusion emerge from the results.

*Structured generation evaluation gap.* Models systematically perform better at evaluating explanations than generating them, especially for tasks requiring structured numerical reasoning.

*Limits of numerical reasoning.* Failures on tasks such as Volatility Shift and anomaly counting indicate that LLMs struggle with higher order statistical reasoning and precise quantitative calibration, even when they perform well on simpler pattern recognition tasks.

## 4.6. Limitations

Our study has several limitations. First, explanation generation outputs are annotated by a single domain-knowledgeable annotator, which may introduce subjective bias, although the structured rubric reduces ambiguity. Second, TSQueryBench consists entirely of synthetic time-series data, enabling controlled evaluation but leaving generalization to real-world data as an open question. Third, LLM-based evaluation may be sensitive to prompt design, which we do not analyze. Finally, although candidate order is randomized in the ranking task, we do not explicitly test for residual positional bias.

## 5. Conclusion

We introduced **TSQueryBench**, a controlled benchmark for evaluating data-grounded reasoning in time-series explanation tasks. Across generation, ranking, scoring, and anomaly detection, we find a consistent generation evaluation asymmetry: models often fail to produce numerically correct explanations while reliably identifying or scoring correct alternatives, especially on tasks requiring structured numerical reasoning. These results indicate that while current LLMs struggle with generating faithful explanations, rubric guided evaluation remains reliable, suggesting the primary bottleneck lies in generation rather than underlying understanding. TSQueryBench provides a foundation for

studying these behaviors and developing more robust evaluation methods, with future work focusing on improving data-grounded generation through prompt design, structured decoding, and advanced reasoning strategies.

## Acknowledgements

The authors acknowledge the use of AI tools such as ChatGPT, Claude, and Gemini for improving the presentation and grammar of this paper. All results, analyses, and proposed methods are solely the authors' contributions. The authors take full responsibility for the content of this paper.

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

## A. Extended Results

This appendix provides detailed quantitative results supporting the findings discussed in Section 4.

### A.1. Explanation Generation

*Table 2.* Explanation generation accuracy per query type. Best per row in **bold**, second-best underlined.

| QUERY TYPE | LLAMA 3.1 8B | GEMMA 4 31B | GEMMA 3 12B | QWEN 3 8B | CLAUDE 3 HAIKU | DEEPSEEK V3.2 |
|---|---|---|---|---|---|---|
| LINEAR SPIKE | 0.70 | **0.98** | 0.46 | 0.94 | 0.68 | 0.94 |
| SEASONAL DROP | 0.06 | 0.12 | 0.00 | 0.16 | **0.24** | 0.08 |
| STRUCTURAL BREAK | 0.46 | 0.52 | 0.04 | 0.18 | **0.68** | 0.46 |
| MULTI-METRIC CONSISTENCY | **0.18** | 0.12 | 0.00 | 0.14 | 0.10 | 0.00 |
| RELATIVE EXTREMUM | 0.12 | **0.60** | 0.08 | 0.02 | 0.32 | 0.10 |
| MEAN SHIFT | 0.02 | 0.00 | 0.02 | **0.88** | 0.02 | 0.08 |
| VOLATILITY SHIFT | 0.00 | **0.04** | 0.02 | 0.00 | 0.00 | 0.00 |
| TREND COMPARISON | 0.00 | 0.28 | 0.04 | 0.67 | 0.42 | **0.70** |
| TEMPORAL ORDERING | 0.12 | 0.08 | 0.22 | **0.26** | 0.14 | 0.12 |
| SHAPE CLASSIFICATION | 0.18 | **0.84** | 0.38 | 0.60 | 0.48 | 0.54 |

*Table 3.* Overall error distribution for explanation generation. Highest per column in **bold**, second-highest underlined.

| MODEL | WRONG | PARTIAL | CORRECT |
|---|---|---|---|
| QWEN 3 8B | 0.39 | 0.25 | **0.36** |
| GEMMA 4 31B | 0.41 | 0.23 | **0.36** |
| DEEPSEEK V3.2 | 0.36 | **0.34** | 0.30 |
| CLAUDE 3 HAIKU | 0.43 | 0.26 | 0.31 |
| LLAMA 3.1 8B | 0.59 | 0.22 | 0.18 |
| GEMMA 3 12B | **0.64** | 0.24 | 0.13 |

### A.2. Relative Ranking

*Table 4.* Relative ranking accuracy per query type. Best per row in **bold**, second-best underlined.

| QUERY TYPE | LLAMA 3.1 8B | GEMMA 4 31B | GEMMA 3 12B | QWEN 3 8B | CLAUDE 3 HAIKU | DEEPSEEK V3.2 |
|---|---|---|---|---|---|---|
| LINEAR SPIKE | 0.60 | 0.50 | 0.50 | **0.96** | 0.60 | 0.54 |
| SEASONAL DROP | 0.42 | 0.52 | 0.20 | **0.92** | 0.18 | 0.48 |
| STRUCTURAL BREAK | 0.58 | **0.94** | 0.80 | **0.94** | 0.38 | 0.86 |
| MULTI-METRIC CONSISTENCY | 0.48 | 0.54 | 0.54 | **0.94** | 0.50 | 0.58 |
| RELATIVE EXTREMUM | 0.32 | 0.62 | 0.50 | 0.38 | 0.22 | **0.64** |
| MEAN SHIFT | 0.28 | 0.72 | 0.60 | **0.94** | 0.22 | 0.84 |
| VOLATILITY SHIFT | 0.46 | 0.44 | 0.52 | **0.62** | 0.36 | 0.54 |
| TREND COMPARISON | 0.76 | **0.98** | 0.58 | 0.96 | 0.56 | 0.94 |
| TEMPORAL ORDERING | 0.56 | 0.44 | 0.85 | 0.76 | 0.58 | **0.98** |
| SHAPE CLASSIFICATION | 0.48 | **1.00** | 0.48 | 0.98 | 0.48 | **1.00** |

*Table 5.* Ranking accuracy by time-series length. Best per row in **bold**, second-best underlined.

| LENGTH | LLAMA 3.1 8B | GEMMA 4 31B | GEMMA 3 12B | QWEN 3 8B | CLAUDE 3 HAIKU | DEEPSEEK V3.2 |
|---|---|---|---|---|---|---|
| 100 | 0.48 | 0.69 | 0.53 | **0.87** | 0.39 | 0.79 |
| 200 | 0.44 | 0.70 | 0.58 | **0.86** | 0.46 | 0.73 |
| 300 | 0.56 | 0.66 | 0.48 | **0.81** | 0.46 | 0.69 |
| 500 | 0.51 | 0.61 | 0.44 | **0.79** | 0.34 | 0.71 |

## A.3. Independent Scoring

*Table 6.* Independent scoring accuracy per query type. Best per row in **bold**, second-best underlined.

| QUERY TYPE | LLAMA 3.1 8B | GEMMA 4 31B | GEMMA 3 12B | QWEN 3 8B | CLAUDE 3 HAIKU | DEEPSEEK V3.2 |
|---|---|---|---|---|---|---|
| LINEAR SPIKE | 0.35 | 0.67 | 0.59 | **0.96** | 0.40 | 0.62 |
| SEASONAL DROP | 0.27 | 0.68 | 0.51 | **0.82** | 0.28 | 0.42 |
| STRUCTURAL BREAK | 0.22 | **0.95** | 0.63 | 0.75 | 0.39 | 0.85 |
| MULTI-METRIC CONSISTENCY | 0.38 | **0.67** | 0.33 | 0.65 | 0.31 | 0.33 |
| RELATIVE EXTREMUM | 0.16 | **0.61** | 0.15 | 0.45 | 0.43 | 0.31 |
| MEAN SHIFT | 0.27 | 0.71 | 0.36 | **0.91** | 0.33 | 0.79 |
| VOLATILITY SHIFT | 0.35 | 0.57 | 0.44 | **0.72** | 0.35 | 0.51 |
| TREND COMPARISON | 0.33 | 0.55 | 0.43 | **0.84** | 0.33 | 0.68 |
| TEMPORAL ORDERING | 0.31 | 0.41 | **0.61** | 0.47 | 0.33 | 0.47 |
| SHAPE CLASSIFICATION | 0.29 | **0.90** | 0.61 | 0.67 | 0.33 | 0.58 |

*Table 7.* Scoring accuracy by time-series length. Best per row in **bold**, second-best underlined.

| LENGTH | LLAMA 3.1 8B | GEMMA 4 31B | GEMMA 3 12B | QWEN 3 8B | CLAUDE 3 HAIKU | DEEPSEEK V3.2 |
|---|---|---|---|---|---|---|
| 100 | 0.29 | 0.70 | 0.49 | **0.72** | 0.35 | 0.57 |
| 200 | 0.25 | 0.68 | 0.47 | **0.74** | 0.35 | 0.56 |
| 300 | 0.24 | 0.69 | 0.66 | **0.74** | 0.35 | 0.54 |
| 500 | 0.32 | 0.61 | 0.68 | **0.70** | 0.35 | 0.54 |

## A.4. Multi-Anomaly Detection

*Table 8.* Multi-anomaly detection performance (Count Accuracy and F1 Score). Best per row in **bold**, second-best underlined.

| | LLAMA 3.1 8B | | GEMMA 4 31B | | GEMMA 3 12B | | QWEN 3 8B | | CLAUDE 3 HAIKU | | DEEPSEEK V3.2 | |
|---|---|---|---|---|---|---|---|---|---|---|---|---|
| LENGTH | CA | F1 | CA | F1 | CA | F1 | CA | F1 | CA | F1 | CA | F1 |
| 100 | 0.00 | 0.414 | **0.56** | **0.991** | 0.00 | 0.528 | 0.08 | 0.500 | 0.00 | 0.572 | 0.20 | 0.932 |
| 200 | 0.04 | 0.391 | **0.60** | **0.976** | 0.00 | 0.270 | 0.12 | 0.455 | 0.00 | 0.492 | 0.48 | 0.871 |
| 300 | 0.00 | 0.337 | **0.24** | **0.910** | 0.00 | 0.155 | 0.12 | 0.585 | 0.00 | 0.700 | 0.16 | 0.853 |
| 500 | 0.00 | 0.205 | **0.40** | **0.632** | 0.00 | 0.148 | 0.16 | 0.576 | 0.04 | 0.696 | 0.16 | 0.838 |
| AVG. | 0.01 | 0.337 | **0.45** | **0.877** | 0.00 | 0.275 | 0.12 | 0.529 | 0.01 | 0.615 | 0.25 | 0.874 |

# B. Dataset Samples and Evaluation Traces

This appendix presents a complete end-to-end walkthrough of our evaluation framework, followed by one representative example per task drawn from TSQueryBench and the multi-anomaly dataset. All model outputs are from Qwen 3 8B in a zero-shot setting. Label encoding: 0 = Incorrect, 1 = Partially Correct, 2 = Fully Correct.

## B.1. End-to-End Illustrative Example

We walk through a single instance from the Linear Spike category to illustrate how all four tasks operate on the same underlying data.

**Time series and question.** Figure 1 shows a 45-step window of instance `ls_0`, a linearly trending series with a sharp transient spike at $t=35$ (value 89.20, versus 16.17 at $t=34$) before immediately returning to the underlying trend. The question posed to all models is:

"*Is there an anomaly? If yes, identify when and quantify the change.*"

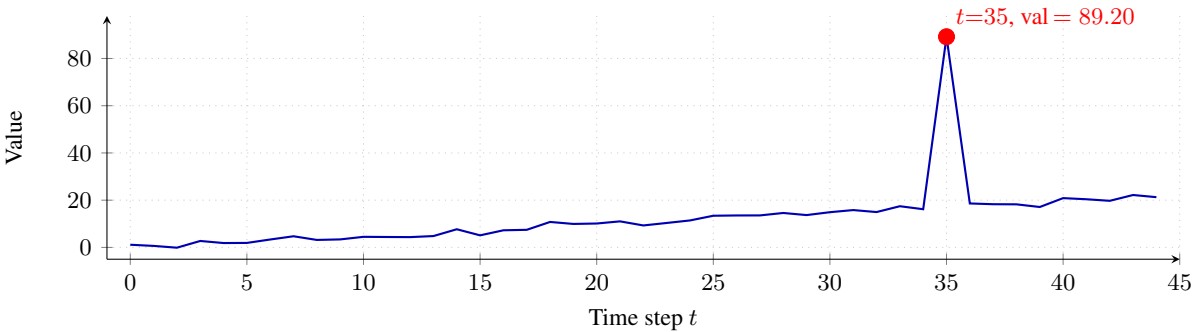

*Figure 1.* Instance `ls_0` (Linear Spike, $n=100$, first 45 steps shown). The series follows a linear upward trend with Gaussian noise. A single transient spike at $t=35$ (•) deviates sharply from the local trend before reverting immediately.

**Candidate explanations.** Three candidates are constructed to represent the three correctness levels:

- **Incorrect** (label 0): "A comprehensive review of the series suggests no statistically significant anomaly. ... The most notable deviation appears around $t=40$, with an estimated increase of roughly $349.93\%$." *[wrong location, denies anomaly]*

- **Partially correct** (label 1): "There is an anomaly at $t=35$. The value increases by approximately $470.4\%$ compared to the previous point. This corresponds to a z-score of about $3.45$." *[correct location, numeric error: true change $\approx 451.5\%$]*

- **Correct** (label 2): "There is a clear anomaly at $t=35$. The value increases by approximately $451.51\%$ compared to the previous point. This corresponds to a z-score of about $4.02$, indicating statistical significance." *[correct location and magnitude]*

**Experiment 1: Explanation generation.** Given only the raw series and question, Qwen 3 8B produces:

*"Yes, there is an anomaly at index 35 with a value of 89.2. This represents a $451.6\%$ increase from the previous value (16.17 at index 34) and is followed by a return to lower values, indicating a temporary sharp deviation."*

The explanation correctly identifies location, direction, and magnitude and is assessed as **Correct** (label 2) by human annotators.

**Experiment 2: Relative ranking.** Presented with all three candidates, the model assigns scores {C0: label 1, C1: label 0, C2: label 2} and selects **C2** as the best explanation, matching the ground truth.

**Experiment 3: Independent scoring.** Each candidate evaluated in isolation:

| Candidate | Predicted | Ground truth |
|---|---|---|
| C0 — partially correct | 1 | 1 |
| C1 — incorrect | 0 | 0 |
| C2 — correct | 2 | 2 |

All three labels are predicted correctly, showing that rubric-guided evaluation reliably distinguishes among correctness levels even in isolation.

**B.2. Explanation Generation: Failure Case**

**Instance.** `vs_2` under Volatility Shift, $n=100$.

**Question:**

*Does the volatility of the series change at any point? If yes, identify approximately when and compare the variability before and after.*

**Ground truth:** Variance reduction around $t=45$; std. dev. decreases from $\approx 7.17$ to $\approx 3.25$.

**Model-generated explanation:**

*"Yes, the volatility decreases after index 34. Before this point, the series exhibited larger swings (e.g., from $-10.72$ at index 8 to $18.63$ at index 20), while after index 34 values fluctuate within a narrower range (e.g., $1.02$ at index 39 to $19.96$ at index 98), indicating reduced variability."*

**Outcome: Incorrect** (label 0). The model detects the qualitative pattern (variance reduction) but mislocates the change point ($t=34$ vs. ground truth $t=45$) and reports no quantitative standard deviation values, failing the numeric accuracy criterion. This is representative of the systematic failure mode on Volatility Shift queries.

### B.3. Relative Ranking: Generation - Evaluation Asymmetry

**Instance.** `vs_2` under Volatility Shift, $n=100$ (same instance as Appendix A.1).

**Candidates and ranking result:**

C1. *"The volatility changes around $t=45$. Std. dev. before: $\approx 7.17$; after: $\approx 3.25$. The first segment exhibits higher variability."* GT: **2**

C2. *"The volatility shifts around $t=45$. Std. dev. before: $\approx 7.0$; after: $\approx 3.7$. The first half shows greater variability."* GT: **1**

C3. *"The series displays uniform variability throughout. Std. dev. $\approx 10.01$ throughout. No structural change is evident; the data appears homoscedastic overall."* GT: **0**

The model assigns {C1: label 2, C2: label 1, C3: label 0} and selects **C1**, matching the ground truth. Although the same model cannot generate a correct explanation for this query type (Appendix A.1), it successfully identifies the most accurate candidate comparatively, directly illustrating the generation-evaluation asymmetry discussed in Section 4.

### B.4. Independent Scoring: Rubric Traces

**Instance.** `sb_0` under Structural Break, $n=100$.

**Question:**

*Is there a structural change? If yes, quantify the shift.*

| Candidate (truncated) | GT | Pred | Correct |
|---|---|---|---|
| "Structural break at $t=50$. Mean shifts from $\approx 9.97$ to $\approx 20.0$, a 100.62% increase." | 2 | 2 | ✓ |
| "Structural change at $t=50$. Mean moves from 8.66 to 21.37, roughly 109.69%." | 1 | 1 | ✓ |
| "Series remains broadly consistent … average level $\approx$ 6.07 throughout … regime stability rather than structural change." | 0 | 0 | ✓ |

**Rubric traces:**

- **C1 → label 2:** Break correctly identified at $t=50$; pre-break mean (9.97) matches ground truth; percentage increase (100.62%) is numerically accurate; no unsupported claims. All rubric dimensions pass.

- **C2 → label 1:** Location correct, but pre-break mean (8.66) and percentage change (109.69%) deviate from verified values. Numeric accuracy dimension fails.

- **C3 → label 0:** Denies any structural change, directly contradicting the observable mean shift. Data faithfulness dimension fails outright.

### B.5. Multi-Anomaly Detection

**Instance.** `multi_1` under Multi-Anomaly, $n{=}100$, 9 ground-truth anomalies.

**Question:**

> *How many anomalies are present in this series? List the index of each anomaly and quantify the percentage change at each one.*

**Ground-truth anomalies:**

| Index | % change | z-score |
|---|---|---|
| 10 | 843.04 | 2.50 |
| 18 | 862.16 | 3.24 |
| 26 | 489.01 | 2.40 |
| 34 | 297.78 | 2.65 |
| 42 | 314.78 | 3.50 |
| 57 | 255.81 | 2.98 |
| 66 | 202.04 | 3.13 |
| 76 | 185.23 | 3.49 |
| 86 | 117.07 | 3.00 |

**Model prediction:** 11 anomalies at indices $\{4, 6, 10, 18, 26, 34, 42, 57, 66, 76, 86\}$.

| | Count correct | Precision | Recall / F1 |
|---|---|---|---|
| `multi_1` | ✗ (pred 11, GT 9) | 0.818 | 1.00 / 0.90 |

All 9 ground-truth anomalies are correctly localised (recall $= 1.0$), but 2 false positives are introduced at $t{=}4$ and $t{=}6$, corresponding to minor boundary fluctuations that do not reach the anomaly threshold. This over-detection pattern of high recall at the cost of precision is consistent with the broader finding that models tend toward over-sensitive detection when no explicit threshold is provided.

## C. Candidate Explanation Construction

Each instance in **TSQueryBench** is paired with three candidate explanations corresponding to distinct correctness levels: fully correct (label 2), partially correct (label 1), and incorrect (label 0). Candidates are generated deterministically from ground-truth statistics via a rule-based pipeline, ensuring that correctness distinctions are controlled and reproducible rather than subject to annotator variability.

**Fully correct explanations (label 2).** The faithful candidate is constructed by directly substituting verified ground-truth values into a fixed natural-language template for each query type. For example, a *Linear Spike* instance yields a statement of the form: *"There is a clear anomaly at $t = k$. The value increases by approximately $p\%$ compared to the previous point, corresponding to a z-score of $z$"*, where $k$, $p$, and $z$ are taken directly from ground-truth statistics. All numerical claims in the faithful candidate are thus guaranteed to be accurate.

**Partially correct explanations (label 1).** The numeric-error candidate preserves the correct qualitative reasoning and identifies the right location or pattern, but introduces controlled perturbations to numerical values. Each scalar quantity $v$ is replaced by $\hat{v} = v \cdot (1 + \epsilon)$, where $\epsilon \sim \text{Uniform}(-0.15, 0.15)$ (i.e., a $\pm 15\%$ perturbation), rounded to two decimal places. This produces explanations that are directionally correct but numerically inaccurate, targeting the *Numeric Accuracy* dimension of the rubric.

**Incorrect explanations (label 0).** The incorrect candidate is designed to be plausible in surface form while being factually inconsistent with the data. Depending on query type, this is achieved through one or more of the following strategies: (i) **denial**, where the model explicitly asserts that no anomaly, shift, or pattern exists; (ii) **index shift**, where the event is attributed to a wrong location (e.g., $t + 4$ or $t + 5$ instead of the true index); (iii) **wrong label**, where an incorrect category is asserted (e.g., classifying a *concave* series as *monotone_increase*); and (iv) **direction reversal**, where the wrong segment or

ordering is identified as dominant. Incorrect explanations additionally employ verbose, hedged language to mimic plausible but unsupported reasoning.

**Randomization and answer key.** For each instance, the three candidates are assigned labels $\{0, 1, 2\}$ and their presentation order is randomized using a fixed random seed (`seed=42`) to eliminate positional bias during evaluation. The shuffled candidates are written to `randomized_dataset.jsonl` and the corresponding label order is stored separately in `answer_key.jsonl`, which is withheld from models at inference time.

**Query-type-specific construction.** Table 9 summarizes the construction strategy for each of the ten query types, specifying which ground-truth fields are used for the faithful candidate and what perturbation or distractor strategy is applied for the incorrect candidate.

*Table 9.* Candidate construction strategy per query type. The "Faithful fields" column lists the ground-truth keys substituted into the correct template. The "Incorrect strategy" column describes how the label-0 candidate is constructed.

| Query Type | Faithful fields | Incorrect strategy |
|---|---|---|
| Linear Spike | `anomaly_index`, `percent_change`, `z_score` | Denial + index shift ($+5$) + large perturbation ($\pm 50\%$) |
| Seasonal Drop | `anomaly_index`, `percent_change`, `z_score` | Denial of drop; attributes trough to seasonal cycle |
| Structural Break | `break_index`, `mean_before`, `mean_after`, `percent_shift` | Denial; reports stable mean with large perturbation ($\pm 50\%$) |
| Multi-Metric | `anomaly_index`, `percent_change`, `z_score`, `variance_before/after` | Denial + fixed small change ($5\%$) + inverted variance direction |
| Relative Extremum | `largest_spike_index`, `largest_spike_magnitude` | Attributes maximum to a different spike index |
| Mean Shift | `mean_first_half`, `mean_second_half`, `percent_difference` | Asserts equal means; uses perturbed first-half mean for both halves |
| Volatility Shift | `break_index`, `std_before`, `std_after`, `higher_volatility_segment` | Asserts homoscedasticity; attributes lower std to higher-variance segment |
| Trend Comparison | `slope_first_half`, `slope_second_half`, `steeper_half` | Asserts equal slopes; uses perturbed first-half slope for both |
| Temporal Ordering | `correct_order`, `segment_means` | Rotated order (cyclic shift by 1 position) |
| Shape Classification | `shape`, `turning_point_index`, `value_at_start/end` | Fixed wrong label (e.g., *concave* $\rightarrow$ *monotone_increase*) |

# D. Related Work

**Reference-based evaluation.** Automatic evaluation of generated text has traditionally relied on reference-based metrics such as BLEU, ROUGE, and embedding-based similarity measures (Papineni et al., 2002; Lin, 2004; Zhang et al., 2020). These metrics quantify lexical or semantic overlap between a generated output and a gold reference, and are widely used in machine translation, summarization, and question answering. Natural language inference (NLI) models similarly assess entailment between candidate outputs and reference statements (Bowman et al., 2015). However, these approaches assume the availability of ground-truth explanations and operate purely in a text-to-text setting. They do not incorporate structured numerical evidence and therefore cannot verify whether an explanation is faithful to underlying time-series data, particularly when numerical inconsistencies are subtle or not reflected in surface-level text similarity.

**Time-series and language.** Classical time-series analysis provides principled methods for modeling temporal dynamics, including trend detection, anomaly identification, and forecasting (Hamilton, 1994; Box et al., 2015). More recently, large language models have been integrated into time-series pipelines, either by reprogramming pretrained LLMs for forecasting or by generating natural-language interpretations of temporal signals (Jin et al., 2023; Zhou & Yu, 2024). Benchmarks such as TSAQA further emphasize explanation-level reasoning over time-series data (Jing et al., 2026). However, evaluation in

these settings typically focuses on forecasting accuracy or relies on surface-level textual similarity, leaving open the problem of systematically verifying whether generated explanations are numerically faithful to the underlying data.

**LLMs as evaluators.**    Recent work has explored using large language models as evaluators of generated outputs. Methods such as GPTScore, G-Eval, and MT-Bench demonstrate that LLMs can approximate human judgments under structured prompting and rubric-based scoring (Fu et al., 2023; Liu et al., 2023; Zheng et al., 2023). Platforms such as Chatbot Arena study comparative evaluation and model calibration (Chiang et al., 2024), while reference-free approaches such as RAGAS highlight the feasibility of model-based evaluation without ground-truth references (Akin et al., 2023). However, these approaches primarily operate in text-to-text settings. The applicability of LLM-based evaluators to numerically grounded domains, where correctness depends on reasoning over structured data rather than textual similarity, remains underexplored.

**Positioning of our work.**    Our work extends the LLM-as-a-Judge paradigm to a data-grounded, reference-free setting for time-series explanations. Unlike reference-based metrics, our approach conditions directly on raw numerical inputs; unlike classical time-series methods, it evaluates free-form textual reasoning. Through TSQueryBench, we introduce a controlled benchmark and a multi-task evaluation framework that enables systematic analysis of both generation and evaluation behavior, with a focus on numerical faithfulness, partial correctness, and structured reasoning failures.

