# OpenReview forum: "TSQueryBench: LLM-as-a-Judge for Time-Series Explanations"
_ICML.cc/2026/Workshop/FMSD — FMSD @ ICML 2026 Poster_

### Official Review · Reviewer_duzj · 2026-05-20
**A well-motivated and timely synthetic benchmark**

**Rating:** 9
**Confidence:** 4

**Review:**

This paper introduces TSQueryBench, a well-motivated and timely synthetic benchmark designed to evaluate the viability of using LLMs as judges for natural language explanations of time-series data. The authors evaluate six LLMs across four distinct tasks—generation, relative ranking, independent scoring, and multi-anomaly detection—uncovering a compelling "generation-evaluation asymmetry" where models reliably score or rank correct explanations better than they can generate them, even on challenging tasks like volatility shifts. The experimental design is rigorous, utilizing a structured ternary rubric and directly conditioning the models on raw numerical data rather than relying on standard text-based reference metrics. However, the study is somewhat limited by its exclusive reliance on synthetic datasets, which leaves its generalizability to noisy, real-world time-series data an open question, and the use of a single human annotator for the generation task introduces potential subjective bias. Overall, the paper provides valuable empirical insights into the numerical reasoning capabilities and bottlenecks of current foundation models.

---

### Official Review · Reviewer_tiLd · 2026-05-21
**A reproducible benchmark hindered by a non-commensurable headline metric and uncited prior work.**

**Rating:** 4
**Confidence:** 4

**Review:**

**Summary:**

The paper introduces TSQueryBench, a synthetic benchmark to evaluate LLMs as judges of natural-language explanations for time-series data. Spanning 500 instances and evaluating six LLMs across four tasks, the paper claims a "generation–evaluation asymmetry," where models that fail to generate correct explanations can still reliably rank and score them.

**Strengths:**

- The benchmark-construction pipeline is highly reproducible, deterministic, and removes annotator variance from the correctness labels.
- The four-task decomposition provides a coherent framework to probe generative versus evaluative capabilities.
- The inclusion of end-to-end traces in the appendices allows for excellent auditing of individual judgments.

**Weaknesses:**

- The central claim of a "generation–evaluation asymmetry" relies on comparing non-commensurable quantities (a strict generation quality-rate with a floor near zero versus a 3-way classification metric with a 0.33 chance floor), lacking chance correction or confidence intervals.
- The benchmark suffers from a surface-cue confound; incorrect candidates can be identified via systematic tells without actual numerical verification against the series.
- The claim that "LLM judges are reliable" is contradicted by the paper's own tables, where multiple models perform at or below the 3-way chance floor.
- An incorrect novelty claim: a near-identical prior version by the authors (arXiv:2604.02118) is uncited, and the asymmetry phenomenon is already well-established in LLM literature.

---

### Official Review · Reviewer_zQ13 · 2026-05-22
**A clear time series explanations LLM evaluation benchmark**

**Rating:** 6
**Confidence:** 4

**Review:**

This paper introduces TSQueryBench, a synthetic benchmark with 500 underlying time-series. The paper studies how well LLMs (six mixed family/size studied (<=32B)) can verify numerical and logical correctness of the explanations through 4 different experiments (Explanation Generation, Relative Ranking, Independent Scoring, Multi-Anomaly Detection). The main finding is that LLMs can be good TS explanation judges even if they are weak generators (generation-evaluation asymmetry).

Strengths:

- Clear paper, interesting main finding
- The paper demonstrates that while text generation degrades or shifts based on output constraints, rubric-guided evaluation performance remains remarkably stable across time-series lengths from 100 to 500 timesteps.
- Hard negative evaluation adds to the strength of the claims
- Rubric style evaluation is appreciated
- Code is supplied

Areas for Improvement:

- Missing the human eval possible upper bound. Hard to place in context now. Even only on one main experiment would have been appreciated.
- Missing small models (e.g. 1B variants) to define a lower bound.
- The authors failed to provide inference time, throughput, or computational cost data.
- In Experiment 4, the models show low count acc. but high F1 scores, showing over-reporting of anomalies. If an LLM judge is fundamentally prone to hallucinating anomalies, its baseline capability as a reliable judge is somewhat compromised. It risks penalizing accurate explanations for failing to mention "anomalies" that are actually just minor noise artifacts.
- Real-world time-series data is rarely homoscedastic. If an LLM struggles with synthetic (heteroscedastic) volatility shifts, it seems that its evaluative reliability is likely to break down in messy real-world scenarios.
- Candidates explanations follow a few rigid structures. Although this supports the approach of testing numerical evaluation, the numerical evaluation itself might be affected by the wording around it. This remains unexplored.
- The paper does not provide a clear description of the data generation process for the underlying synthetic time-series signals themselves (even though code is available).
- No multivariate study
- Code: Exp1 evaluation script is missing (e1_eval_helper_b.py is empty).
- Not clear what sequence length (or aggregation) is used for table 1,2,4, and 6.
- Example TS graph per query type would be appreciated.

Justification of Score
Despite limitations regarding synthetic data constraints, a lack of multivariate testing, and missing compute metrics, this paper provides a highly clear, well-structured, and interesting study on the evaluative boundaries of LLMs. The documentation of the generation-evaluation asymmetry offers valuable insights that directly align with the workshop's focus on benchmarking alternative LLM paradigms for structured data. Addressing the missing cost and latency data will elevate this to a strong contribution.